# The active layer soils of Greenlandic permafrost areas can function as important sinks for volatile organic compounds
Yi Jiao [1,2] ✉, Magnus Kramshøj[2], Cleo L. Davie-Martin[2,3], Bo Elberling [4] & Riikka Rinnan [1,2] ✉

Permafrost is a considerable carbon reservoir harboring up to 1700 petagrams of carbon accumulated over millennia, which can be mobilized as permafrost thaws under global warming. Recent studies have highlighted that a fraction of this carbon can be transformed to atmospheric volatile organic compounds, which can affect the atmospheric oxidizing capacity and contribute to the formation of secondary organic aerosols. In this study, active layer soils from the seasonally unfrozen layer above the permafrost were collected from two distinct locations of the Greenlandic permafrost and incubated to explore their roles in the soil-atmosphere exchange of volatile organic compounds. Results show that these soils can actively function as sinks of these compounds, despite their different physiochemical properties. Upper active layer possessed relatively higher uptake capacities; factors including soil moisture, organic matter, and microbial biomass carbon were identified as the main factors correlating with the uptake rates. Additionally, uptake coefficients for several compounds were calculated for their potential use in future model development. Correlation analysis and the varying coefficients indicate that the sink was likely biotic. The development of a deeper active layer under climate change may enhance the sink capacity and reduce the net emissions of volatile organic compounds from permafrost thaw.

Permafrost is ground that remains frozen for at least two consecutive years and underlies about 11% of the global surface[1]. Its frozen state preserves substantial amounts of dead organic matter, which has been immobilized over millennia without the opportunity to fully decompose. The permafrost carbon reservoir is estimated to harbor up to 1700 petagrams (1 Pg = $10^{15}$ g) of carbon—as much as twice the carbon content of the atmosphere[2]. With the rise in global temperatures, permafrost thaw is increasingly evident, mobilizing ancient organic matter. Upon thaw, part of the permafrost carbon is released into the atmosphere as greenhouse gases, such as carbon dioxide and methane[3,4]. Recently, the release of volatile organic carbon (VOC) from thawing permafrost has also been observed, and this is garnering increasing research attention[5–8]. Emissions of these VOCs contribute to the atmospheric carbon dioxide reservoir, because they are eventually oxidized to carbon dioxide ($CO_2$). More importantly, they can impact atmospheric chemistry and climate through their reactions with hydroxyl (•OH) radicals[9] and their role in the formation of secondary organic aerosols[10].

Permafrost is overlain by an active layer, which experiences seasonal freeze-thaw cycles, provides a rooting zone for plants, and functions as a seasonal aquifer for near-surface groundwater[11]. Its depth generally ranges from 40 to 320 cm[12] and this layer harbors a greater microbial diversity compared to the underlying permafrost zone[13]. The active layer of soil and its microbial communities play pivotal roles in mediating the exchange of greenhouse gases, heat, and moisture between the atmosphere and the permafrost. For example, different types of psychrophilic methanotrophs have been found in the active layer soils of the permafrost areas, thereby possessing significant methane ($CH_4$) oxidation potential and acting as $CH_4$ sinks[14–16]. Some studies have shown that soils may also have the ability to

[1]Center for Volatile Interactions (VOLT), Department of Biology, University of Copenhagen, Universitetsparken 15, Copenhagen, Denmark. [2]Terrestrial Ecology Section, Department of Biology, University of Copenhagen, Universitetsparken 13A, Copenhagen, Denmark. [3]The Climate and Environmental Research Institute (NILU), The Fram Centre, Tromsø, Norway. [4]Department of Geosciences and Natural Resource Management, University of Copenhagen, Øster Voldgade 10, Copenhagen, Denmark. ✉e-mail: jiaoyi.joey@gmail.com; yi.jiao@bio.ku.dk; riikkar@bio.ku.dk

uptake VOCs when they are readily available to the soil microbes[17,18]. Microbial activity in the active layer has thus been suggested to be crucial for controlling net VOC emissions from permafrost[19]. Nevertheless, we still have a limited understanding of the potential role the active layer soils play in VOC uptake and regulating the net exchange rates between the atmosphere and the thawing cryosphere, especially given the heterogeneity of the active layer and future climate conditions.

The distribution of organic matter, microbial communities, and soil environmental conditions exhibit distinct variations along the vertical profile of the active layer[20–22]. These variations may result in different capacities of the active layer to metabolize VOCs at various depths and times of the year. At the same time, the depth and biogeochemical properties of the active layer are increasingly modified by climate change-induced permafrost thaw. It is projected that active layer depth is increasing at a global average rate of $2.56 \pm 0.07$ cm decade$^{-1}$ under the intermediate carbon emission trajectory (Representative Concentration Pathway 4.5)[12]. The changes in the thickness and properties of the active layer, as well as the warming of the active layer[23], will likely affect biogeochemical processes, including VOC consumption, and thereby the uptake of VOCs diffusing from the permafrost towards the atmosphere. Investigating the VOC consumption rates of the active layer soil at different depths would help us better predict the future role of the deeper active layer in serving as a regulatory layer for VOC emissions.

Greenland is marked by its extensive permafrost coverage, particularly along its coastal areas[24]. In many parts of Greenland, permafrost depths exceed 100 meters, with the upper 1–2 meters comprising either an active layer or permafrost that is vulnerable to degradation under climate change. The retreat of glaciers induced by a warming climate has revealed new surfaces in glacial forefields, and these exposed areas are now subject to primary plant succession and soil formation processes[25]. The emerging soil microbial communities in recently deglaciated terrains significantly differ from those in later successional stages, often characterized by lower biomass and a higher abundance of bacteria[26]. The active layer varies across different locations in age and characteristics, presenting a unique opportunity to study its potential to mediate volatile interactions with the atmosphere.

To assess the role of the active layer of soil in the biogeochemical cycling of VOCs, we collected soil samples from distinct locations in Greenland at different depths. These samples were subjected to controlled laboratory incubations to measure net VOC exchange rates under various environmental conditions. By exposing the soil samples to predetermined VOC mixtures at parts per billion levels, we assessed the potential consumption rates of these compounds in the active layer soils. Particularly, our study was designed to test three hypotheses. First, we expected that the active layer soils possess the capacity for VOC uptake, especially in the upper horizon with the highest microbial biomass and activity. Second, we hypothesized that the older, coastal soils, which likely host more developed, diverse microbial communities than soils close to the ice sheet, would demonstrate a higher and more consistent consumption rate of different VOCs. Third, we expected that the uptake of VOCs is influenced by soil water content (SWC), which will change drastically under the future climate. By testing these hypotheses, this study aims to enhance our understanding of the role of the active layer in the biogeochemistry of VOCs in permafrost regions under climate change.

## Results

### Soil properties

Soils from the two locations exhibited different physicochemical properties ($P < 0.001$, multivariate analysis of variance (MANOVA), Table 1). Specifically, soils from Disko Island were characterized by higher SWC ($P < 0.001$, analysis of variance (ANOVA)), soil organic matter (SOM) ($P < 0.001$, ANOVA), and microbial biomass carbon (Cmic) ($P = 0.006$, ANOVA), whereas soils from Kangerlussuaq contained higher levels of extractable carbon and nitrogen, such as dissolved organic carbon (DOC) ($P = 0.037$, ANOVA) and total dissolved nitrogen (TDN) ($P = 0.017$, ANOVA).

Within each site, significant differences in all soil properties were observed across soil depths (all $P$ values < 0.05, ANOVA), with the exception

**Table 1 | Physiochemical properties of the soil samples**

| Site | Depth [cm] | Proximity | SWC [%] | SOM [%] | NO$_3^-$ [µg g$^{-1}$ dw] | PO$_4^-$ [µg g$^{-1}$ dw] | DOC [µg g$^{-1}$ dw] | TDN [µg g$^{-1}$ dw] | Cmic [µg g$^{-1}$ dw] | Nmic [µg g$^{-1}$ dw] | Pmic [µg g$^{-1}$ dw] | TC [%] | TN [%] |
|---|---|---|---|---|---|---|---|---|---|---|---|---|---|
| Disko Island | 10 | NA | 24.3 ± 2.0 | 11.4 ± 0.7 | 0.6 ± 0.1 | 1.7 ± 0.2 | 152.2 ± 9.0 | 12.1 ± 1.0 | 293.0 ± 69.6 | 13.9 ± 4.3 | 0.7 ± 0.4 | 4.5 ± 0.7 | 0.2 ± 0.0 |
| | 20 | NA | 23.2 ± 3.0 | 10.4 ± 0.7 | 0.8 ± 0.2 | 2.0 ± 0.4 | 177.8 ± 26.9 | 13.0 ± 1.7 | 415.3 ± 116.7 | 29.5 ± 6.4 | 2.2 ± 0.8 | 3.6 ± 0.5 | 0.2 ± 0.0 |
| | 40 | NA | 10.7 ± 0.8 | 7.3 ± 0.3 | 0.2 ± 0.0 | 0.7 ± 0.1 | 58.9 ± 14.2 | 4.7 ± 1.2 | 118.0 ± 18.0 | 10.5 ± 1.3 | 1.4 ± 0.3 | 0.8 ± 0.1 | 0.0 ± 0.0 |
| Kangerlussuaq | 10 | Coast | 11.0 ± 0.4 | 5.7 ± 0.9 | 3.1 ± 1.4 | 1.7 ± 0.7 | 107.8 ± 7.3 | 12.5 ± 1.6 | 164.8 ± 49.6 | 28.1 ± 8.8 | 6.1 ± 2.6 | 3.1 ± 0.5 | 0.1 ± 0.0 |
| | | Intermediate | 2.5 ± 0.4 | 1.8 ± 0.1 | 0.5 ± 0.1 | 1.6 ± 0.3 | 89.6 ± 7.3 | 8.7 ± 1.1 | 94.8 ± 16.0 | 28.3 ± 6.3 | 11.6 ± 2.6 | 0.9 ± 0.1 | 0.0 ± 0.0 |
| | | Glacier | 4.8 ± 0.2 | 3.8 ± 0.1 | 0.7 ± 0.1 | 0.6 ± 0.0 | 100.4 ± 5.7 | 9.3 ± 0.6 | 174.8 ± 24.8 | 21.1 ± 3.0 | 3.2 ± 0.3 | 1.7 ± 0.1 | 0.1 ± 0.0 |
| | 20 | Coast | 4.5 ± 1.1 | 2.2 ± 1.3 | 0.2 ± 0.0 | 0.5 ± 0.1 | 105.7 ± 18.7 | 6.7 ± 1.4 | 17.3 ± 17.3 | 1.3 ± 1.0 | 42.1 ± 41.4 | 1.4 ± 0.9 | 0.1 ± 0.0 |
| | | Intermediate | 1.0 ± 0.0 | 2.1 ± 0.2 | 0.1 ± 0.0 | 0.2 ± 0.0 | 103.4 ± 14.6 | 6.9 ± 1.0 | 38.9 ± 25.5 | 1.4 ± 0.7 | 1.0 ± 0.2 | 0.9 ± 0.1 | 0.1 ± 0.0 |
| | | Glacier | 3.8 ± 0.9 | 2.4 ± 0.3 | 0.5 ± 0.0 | 0.7 ± 0.0 | 111.8 ± 3.1 | 9.3 ± 0.4 | 32.7 ± 27.2 | 6.7 ± 4.7 | 0.9 ± 0.7 | 1.0 ± 0.2 | 0.1 ± 0.0 |
| | 40 | Coast | 3.0 ± 0.4 | 0.5 ± 0.0 | 0.2 ± 0.0 | 0.3 ± 0.0 | 68.6 ± 2.4 | 4.0 ± 0.2 | 14.9 ± 7.5 | 0.8 ± 0.4 | 0.6 ± 0.0 | 0.2 ± 0.0 | 0.0 ± 0.0 |
| | | Intermediate | 0.4 ± 0.2 | 0.9 ± 0.1 | 0.3 ± 0.0 | 0.4 ± 0.0 | 98.7 ± 7.8 | 7.4 ± 0.7 | 6.0 ± 6.0 | 0.0 ± 0.0 | 0.2 ± 0.1 | 0.3 ± 0.1 | 0.0 ± 0.0 |
| | | Glacier | 2.7 ± 0.6 | 1.2 ± 0.0 | 0.5 ± 0.0 | 0.8 ± 0.0 | 113.5 ± 4.7 | 8.7 ± 0.7 | 0.0 ± 0.0 | 0.0 ± 0.0 | 0.0 ± 0.0 | 0.4 ± 0.0 | 0.0 ± 0.0 |

Values reported in the table are mean ± S.E. ($n = 6$ for Disko Island, and $n = 3$ for Kangerlussuaq). *SWC* gravimetric soil water content, *SOM* soil organic matter, *DOC* dissolved organic carbon, *TDN* total dissolved nitrogen, *Cmic* microbial biomass carbon, *Nmic* microbial biomass nitrogen, *Pmic* microbial biomass phosphorus.

of microbial phosphorus (Pmic) content in Disko Island soils ($P = 0.09$, ANOVA). In general, soils from 10 cm depth had relatively higher contents of soil nutrients in comparison to the deeper soils, such as SOM, DOC, TDN, TC, and TN. In Kangerlussuaq, we also compared the soils from different sites from the coast to the proximity of the glacier and significant differences in some soil properties, including SWC ($P < 0.001$, ANOVA), SOM ($P < 0.001$, ANOVA), nitrate ($NO_3^-$) ($P = 0.023$, ANOVA), DOC ($P = 0.031$, ANOVA), TDN ($P = 0.035$, ANOVA), total carbon (TC) ($P < 0.001$, ANOVA), and total nitrogen (TN) ($P < 0.001$, ANOVA), were also found for Kangerlussuaq soils of different proximities to the continental glacier.

## VOC emission blends

The soils were sources of VOCs when measured under VOC-free headspace air. Principal component analysis (PCA) on the fluxes of all individual protonated masses revealed that the soils from the two locations had intrinsically different emission blends (Fig. 1). The first PC, which explained 54.9 % of the variance, differed significantly between the locations ($P < 0.001$, ANOVA). No significant effect of soil depth was observed. However, for Kangerlussuaq soils, there was a significant effect of proximity to the glacier on the VOC profiles ($P < 0.001$, ANOVA). The loadings plot (Fig. 1) suggests that heavier masses (>137 Da) were intrinsically more prevalent in emissions from Kangerlussuaq soils. None of the 12 target compounds studied in the uptake experiments were among the top 10 emitted compounds (their masses and emission rates are provided in Supplementary Table S1). A full mass list of the tentative compounds emitted by these soils, along with their emission rates, is available in the supplementary dataset[27].

## VOC uptake of the active layer soils

In the experiments evaluating VOC uptake (Fig. 2), active layer soils from Disko Island demonstrated the ability to consume all 12 target VOCs, regardless of soil depth, with the exception of acetaldehyde, which was not absorbed by the deeper soils (20 cm and 40 cm depth). In general, there was a trend of 10 cm depth exhibiting higher VOC uptake rates compared to the deeper layers, but this was not statistically significant for compounds other than methanol ($P = 0.008$, Tukey's HSD). The soils displayed particularly high uptake rates for benzyl alcohol, isoprene, and linalool ($-7.1 \pm 0.3$ nmol g dw$^{-1}$ hr$^{-1}$, $-5.8 \pm 0.3$ nmol g dw$^{-1}$ hr$^{-1}$, and $-3.7 \pm 0.2$ nmol g dw$^{-1}$ hr$^{-1}$, respectively, for the 10 cm depth soils).

Active layer soils from Kangerlussuaq, similar to those in Disko Island, demonstrated the capacity to consume all 12 target VOCs

across all depths and proximities to the continental glacier, with the exception of acetaldehyde, which was not absorbed by the deeper soil depths from the coastal site (Fig. 3). Notably, the soils exhibited high uptake rates for isoprene, benzyl alcohol, and linalool, with respective flux rates of $-5.0 \pm 0.1$ nmol g dw$^{-1}$ hr$^{-1}$, $-3.1 \pm 0.1$ nmol g dw$^{-1}$ hr$^{-1}$, and $-1.6 \pm 0.1$ nmol g dw$^{-1}$ hr$^{-1}$, respectively, for the 10 cm depth soils at the coast site.

When comparing the effects of soil depth and location on the overall VOC uptake rates, soil depth did not have a significant effect ($P = 0.14$, MANOVA), whereas the proximity to the glacier exhibited a significant influence ($P < 0.001$, MANOVA). No interaction effect between soil depth and proximity to the glacier was observed. For the individual VOCs, the uptake rates for acetone ($P = 0.003$, ANOVA) and 2-butanone ($P = 0.008$, ANOVA) showed significant differences across different soil depths, with the 10 cm soil depth possessing the highest uptake rates ($P < 0.05$, Tukey's HSD). For other compounds, soil depth did not affect the uptake rate. There was a significant difference between the three locations in the uptake of most VOCs, with the intermediate site showing the smallest uptake rates ($P < 0.05$, Tukey's HSD for compounds other than 2-butanone, toluene, and furfural).

## Water addition effect on VOC uptake rates

Compared to the field-moisture conditions, the addition of water significantly increased uptake rates for all 12 target VOCs ($P < 0.001$, MANOVA, Fig. 4) for all soil samples in Disko Island. For example, the uptake rates for methanol, acetone, and isoprene increased on average by 11.3%, 33.3%, and 23.8%, respectively (all their $P$ values < 0.05, ANOVA). In the case of Kangerlussuaq soils (Fig. 5), moistened soils also showed increased uptake rates compared to their field-moist counterparts, with the exception of acetaldehyde ($P < 0.001$, ANOVA), furfural ($P < 0.001$, ANOVA), and toluene ($P = 0.006$, ANOVA), which displayed contrary trends. Meanwhile, 2-butanone ($P = 0.741$, ANOVA), isoprene ($P = 0.333$, ANOVA), and benzyl alcohol ($P = 0.741$, ANOVA) did not exhibit significant differences in uptake rates between the two moisture levels.

## VOC uptake coefficients

To account for the varying availability of VOCs supplied to the soil consumption, uptake coefficients were calculated by normalizing the uptake rates to the VOC concentrations in the headspace. The averaged uptake coefficients across different soil depths and sites are provided in Table 2. Uptake coefficients for individual soil depths and sites are available in the supplementary dataset[27]. Soils from the two locations displayed significantly

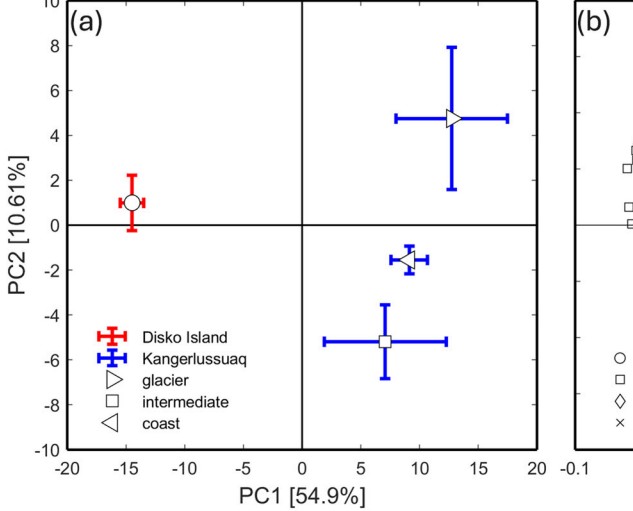
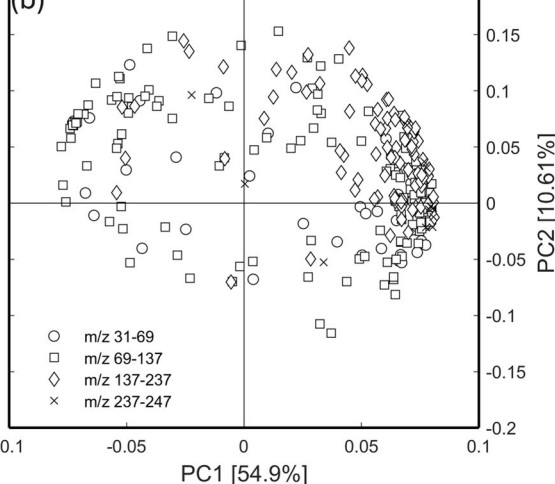
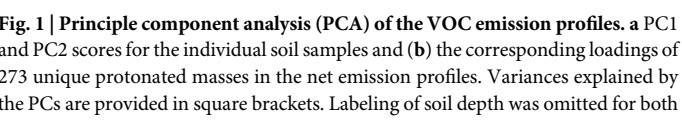

**Fig. 1 | Principle component analysis (PCA) of the VOC emission profiles. a** PC1 and PC2 scores for the individual soil samples and (**b**) the corresponding loadings of 273 unique protonated masses in the net emission profiles. Variances explained by the PCs are provided in square brackets. Labeling of soil depth was omitted for both soils since there was no significant difference. Site information in the score plot is coded by color; the proximities to the glacier for Kangerlussuaq soils (score plot), and m/z range information in the loading plot are coded by symbols.

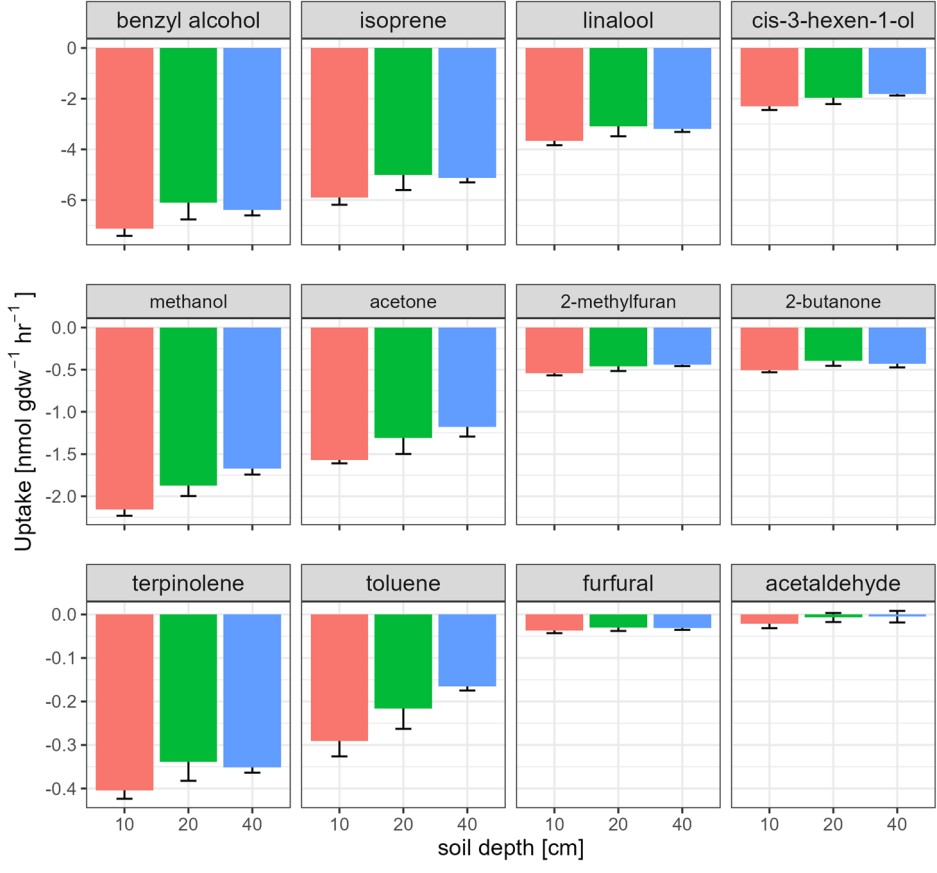

**Fig. 2 | Soil uptake rates of the 12 target VOCs across different soil depths in active layer soil from Disko Island.** The bars represent the mean ± S.E. ($n = 6$). Note different y-axis scales for each row. Positive fluxes represent emissions and negative fluxes uptake. Soil depth had no impact on the overall uptake rates of the 12 VOCs ($P = 0.344$, MANOVA) but it affected the uptake of methanol ($P = 0.008$, ANOVA).

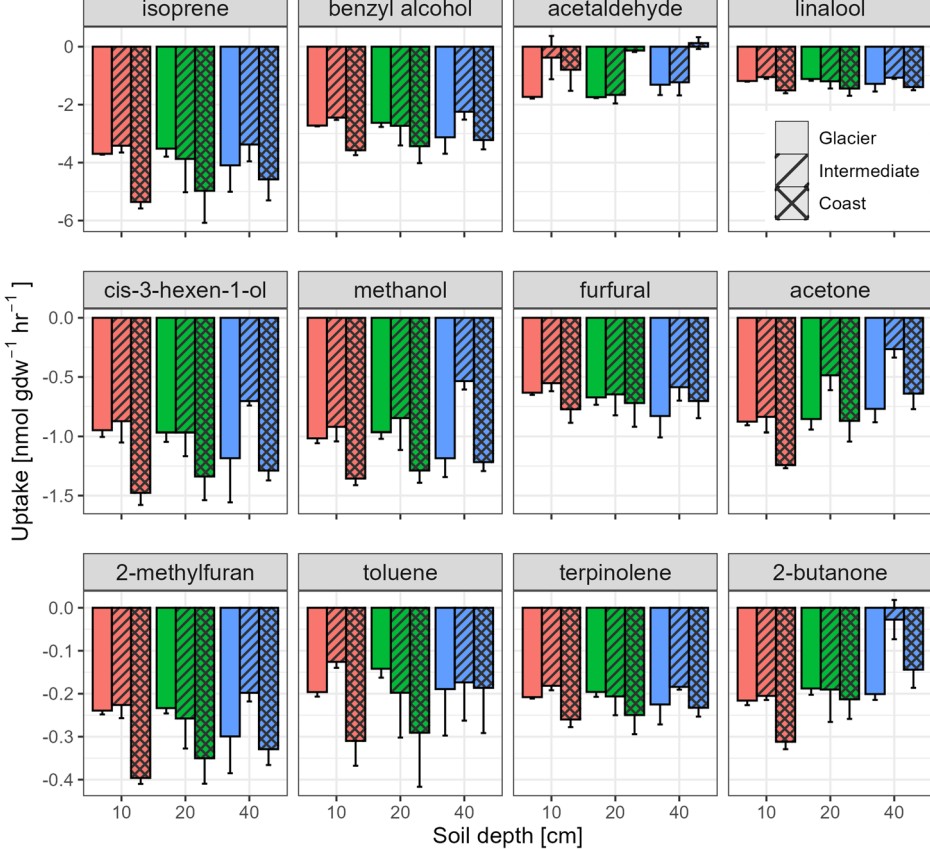

**Fig. 3 | Soil uptake rates of the 12 target VOCs in relation to different depths and proximities to the continental glacier in Kangerlussuaq.** The bars represent the mean ± S.E. ($n = 3$). Note different $y$ axis scales for each row. Positive rates represent emissions and negative rates uptake. Proximity to the glacier affected the overall uptake rates of the 12 VOCs ($P < 0.001$, MANOVA). Soil depth had no effect on the overall uptake rates ($P = 0.14$, MANOVA) but it influenced the uptake of acetone ($P = 0.003$, ANOVA) and 2-butanone ($P = 0.008$, ANOVA).

**Fig. 4 | Comparison between field-moist (D) and moistened (DW) soils from Disko Island in terms of their uptake rates of the 12 target VOCs.** The bars represent the mean ± S.E. ($n = 18$). Note different $y$ axis scales for each row. Water addition affected the overall uptake rates of the VOCs (*** $P < 0.001$, MANOVA). Moistened soils generally showed increased uptake rates compared to their field-moist counterparts (all $P < 0.05$, ANOVA), except for furfural.

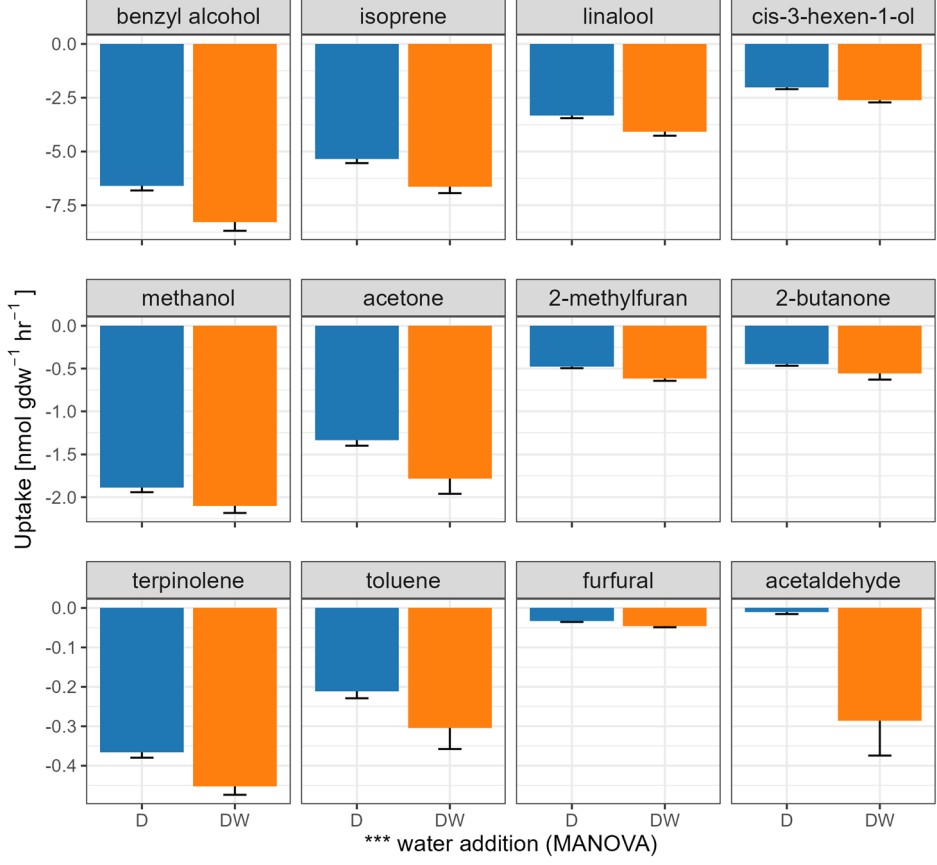

**Fig. 5 | Comparison between field-moist (K) and moistened (KW) soils from Kangerlussuaq in terms of the uptake rates of the 12 target VOCs.** The bars represent the mean ± S.E. ($n = 27$). Note different y-axis scales for each row. Water addition affected the overall uptake rates of the VOCs (*** $P < 0.001$, MANOVA). Moistened soils generally showed increased uptake rates compared to their field-moist counterparts ($P < 0.05$, ANOVA), except for acetaldehyde, furfural, and toluene, which exhibited opposite trends (all $P < 0.05$, ANOVA).

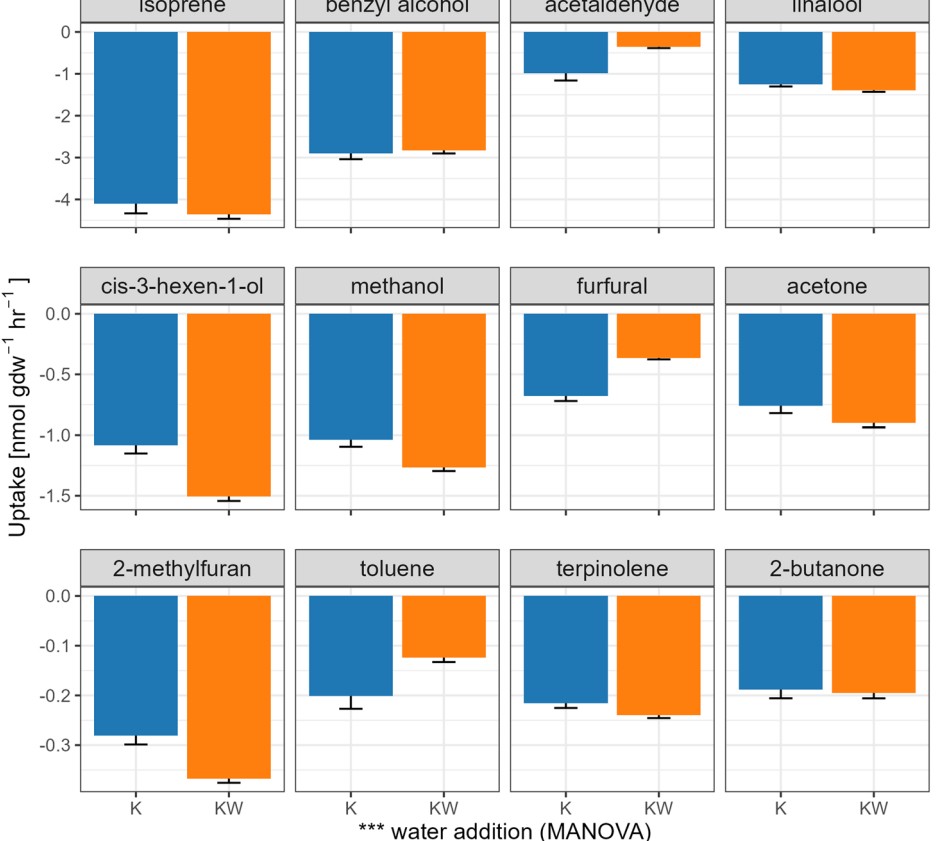

**Table 2 | Averaged uptake coefficients of the 12 target VOCs by the soils from the two locations**

| Compounds | Disko Island | Kangerlussuap |
|---|---|---|
| Methanol | 0.302 ± 0.011 | 0.165 ± 0.009 |
| Acetaldehyde | 0.004 ± 0.002 | 0.383 ± 0.068 |
| Acetone | 0.156 ± 0.009 | 0.088 ± 0.007 |
| Isoprene | 0.078 ± 0.003 | 0.060 ± 0.003 |
| 2-butanone | 0.142 ± 0.008 | 0.060 ± 0.006 |
| 2-methylfuran | 0.196 ± 0.009 | 0.114 ± 0.007 |
| Toluene | 0.053 ± 0.005 | 0.048 ± 0.006 |
| Furfural | 0.009 ± 0.001 | 0.195 ± 0.012 |
| cis-3-Hexen-1-ol | 0.218 ± 0.011 | 0.117 ± 0.007 |
| Benzyl alcohol | 0.304 ± 0.012 | 0.135 ± 0.006 |
| Terpinolene | 0.287 ± 0.013 | 0.170 ± 0.007 |
| Linalool | 0.288 ± 0.013 | 0.108 ± 0.005 |

Values reported in the table are mean ± S.E. ($n = 18$ for Disko Island, and $n = 27$ for Kangerlussuaq).
Unit: mol g dw$^{-1}$ hr$^{-1}$.

different uptake coefficients for the studied VOCs ($P < 0.001$, MANOVA). In general, soils from Disko Island had higher uptake coefficients than those from Kangerlussuaq for most VOCs, including 2-butanone, 2-methylfuran, acetone, benzyl alcohol, cis-3-hexen-1-ol, isoprene, linalool, methanol, and terpinolene ($P < 0.001$, ANOVA). In contrast, soils from Kangerlussuaq had higher uptake coefficients for acetaldehyde and furfural ($P < 0.001$, ANOVA). No difference was found for toluene ($P = 0.541$, ANOVA). On average, the active layer soils showed the highest uptake potentials for methanol, terpinolene, and benzyl alcohol, while 2-butanone, isoprene, and toluene had the lowest uptake potentials.

Soil depth had no significant effect on the uptake coefficients ($P = 0.344$ and 0.149 for Disko Island and Kangerlussuaq, respectively, 0.05, MANOVA). However, significant effects of proximity to glaciers were observed in Kangerlussuaq ($P < 0.001$, MANOVA). Higher uptake coefficients were recorded at the coastal site for 2-butanone, 2-methylfuran, benzyl alcohol, cis-3-hexen-1-ol, and linalool ($P < 0.05$, Tukey's HSD).

## Discussion

Soils can act as sinks for VOCs, and the uptake is closely linked to various physical, chemical, and biological processes[28–30]. Soils have numerous adsorptive sites, such as clay minerals and organic matter, which can bind VOCs. Additionally, soil water can dissolve hydrophilic VOCs and thereby contribute to uptake from the gas phase. Nevertheless, the different uptake coefficients for the different compounds observed in our study suggest that the uptake processes are not merely physical, as more uniform coefficients would have been expected in such cases. Moreover, the uptake coefficients of VOCs did not consistently become higher (i.e., indicating higher uptake potentials) after the water addition. For instance, VOCs such as benzyl alcohol (solubility in water at 20 °C: 40 g L$^{-1}$), furfural (83 g L$^{-1}$), and acetaldehyde (miscible) showed lower uptake potentials after water addition for the Kangerlussuaq soils, despite being water-soluble, which suggests that physical dissolution alone was not the sole driver of the observed VOC uptake in our study. In addition, our earlier study using sterilized fine glass beads as a control for physical adsorption demonstrated that the adsorption rates are typically much lower (<23%) compared to the uptake rates observed with soils[17]. However, glass beads may not fully replicate the complex surface properties of soil particles, and future studies could consider using crushed rocks or other materials to better mimic soil surfaces. Consequently, microbial degradation is likely the primary sink process for VOCs in soils[17,18]. The soil's diverse microbial community is capable of metabolizing a broad spectrum of organic compounds. The observed variation in uptake coefficients for different compounds suggests that microbes in the active layer soils

may exhibit different capacities to degrade, with benzyl alcohol, methanol, and terpinolene appearing to be most favored among the studied VOCs.

The active layer soil's capacity to uptake VOCs and its consistency across the different sites and depths hold significant implications, as these soils may regulate the VOCs released from the thawing permafrost underneath, similar to the process observed in low-affinity methane oxidation. The pan-Arctic region is warming at twice the rate of the global average, resulting in the thawing of permafrost. This thawing mobilizes vast amounts of carbon stored in permafrost[31], releasing significant quantities of VOCs. These include oxygenated compounds, such as methanol and ethanol[19], acetone, and acetaldehyde[6], as well as terpenoid compounds, which exhibit higher chemical reactivity and have profound atmospheric implications[5]. Our experimental setup aimed to replicate natural conditions, where VOCs released from thawing permafrost diffuse upward, accumulate in the soil pore space, and become available for uptake. Our findings suggest that active layer soils are crucial in mediating VOC release from thawing permafrost and thereby, mitigating the associated environmental and climatic impacts. Future studies should consider assessing the effects of increasing active layer thickness, warming, and changes in SWC.

The concentration of gases may be significantly higher in soil pores compared to ambient air if sources exist within the soil medium. For instance, greenhouse gases, such as carbon dioxide ($CO_2$) and methane ($CH_4$), often show elevated concentrations in soil pores. While atmospheric $CO_2$ levels are ~400 parts per million (ppm), soil $CO_2$ concentrations can reach up to 50,000 ppm[32,33]. $CH_4$ concentrations may be lower in the topsoil than in ambient air but are often substantially higher deeper in the soil profile, particularly in anoxic microsites where methanogenesis occurs[34]. Similarly, VOC concentrations are likely to be higher in soils than in ambient air, particularly in soils with high organic matter content or other potential VOC sources. In such cases, the VOC uptake capacity may also be enhanced, as the uptake rate typically follows first-order kinetics, being proportional to substrate concentrations[17,29,35,36]. As a result, relating net VOC concentrations in soil pores to environmental factors becomes challenging[37].

Although the microbial uptake of VOCs in the active layer likely dominates, other processes, such as physical adsorption, dissolution, and chemical degradation, can also contribute. Consequently, the uptake of VOCs by soils is influenced by a combination of factors, including soil physicochemical properties, environmental conditions, and the microbial community and its activity. It is also worth noting that the uptake incubations in this study were relatively short in duration, making it unclear how much physical adsorption or dissolution may have contributed to the observed uptake. Future studies involving longer incubation periods or continuous monitoring of VOC concentrations after stopping the VOC supply would be valuable to identify any potential re-emission and better understand the role of physical processes in VOC uptake.

The distribution of organic matter, microbial communities, and soil environmental conditions often vary significantly along the vertical profile of the active layer[20–22]. Hence, in this study, we measured the soil properties and assessed VOC uptake across different depths along vertical profiles, as well as soils of different ages, as indicated by their proximity to the glacier. Correlation analysis across the different depths and ages revealed that soil moisture, SOM, Total C and N, and particularly microbial biomass carbon, were significantly correlated with the uptake rates of most of the 12 target VOCs (Supplementary Fig. S1), suggesting a strong potential link between VOC uptake and soil microbial activity. As glaciers retreat, they expose new surfaces where plants start to grow, and soil begins to form. Since the primary glacier retreat at the end of the Little Ice Age[38], global warming has sped up this process, uncovering new areas in glacial forefields that undergo early plant succession and soil development[25]. In these recently exposed areas, the soil microbial communities are notably different from those in older soils, with lower biomass and a higher proportion of bacteria compared to fungi[26]. This may explain the higher uptake rates of certain VOCs, such as isoprene, 2-methylfuran, and cis-3-hexen-1-ol, observed in the older coastal soils compared to the younger soils nearer the glacier.

**Table 3 | Concentrations of the 12 target VOCs in the inflow gas stream**

| Compound | Formula | Parent m/z | Concentration [ppbv] |
|---|---|---|---|
| Methanol | $CH_4O$ | 32.04 | 6.30 |
| Acetaldehyde | $C_2H_4O$ | 44.05 | 2.57 |
| Acetone | $C_3H_6O$ | 58.04 | 8.68 |
| Isoprene | $C_5H_8$ | 68.12 | 68.65 |
| 2-butanone | $C_4H_8O$ | 72.11 | 3.14 |
| 2-methylfuran | $C_5H_6O$ | 82.10 | 2.46 |
| Toluene | $C_7H_8$ | 92.14 | 4.20 |
| Furfural | $C_5H_4O_2$ | 96.08 | 3.48 |
| cis−3-Hexen-1-ol | $C_6H_{12}O$ | 100.16 | 9.30 |
| Benzyl alcohol | $C_7H_8O$ | 108.14 | 21.55 |
| Terpinolene | $C_{10}H_{16}$ | 136.23 | 1.27 |
| Linalool | $C_{10}H_{18}O$ | 154.25 | 11.52 |

Soil moisture plays a vital role in regulating VOC production and consumption, thereby influencing net VOC fluxes. It is a key factor driving microbial activity[39], which directly impacts VOC emissions[40–42], enhances microbial uptake of VOCs[40,43], and promotes the dissolution of water-soluble VOCs. However, excessive moisture can limit oxygen diffusion into the soil, potentially slowing aerobic microbial degradation of VOCs. In the active layer soils of the Greenlandic permafrost zone, we observed higher uptake rates for most VOCs following water addition, suggesting moisture-enhanced microbial uptake or increased VOC dissolution. This observation is consistent with findings from an experimental rainforest study[43], where wet soils exhibited net VOC uptake, whereas drought conditions reduced their sink capacity. A long-term drying of the surface active layer soil in permafrost regions is anticipated under future climate conditions[44]. Consequently, while the active layer serves as a critical regulator of VOC transport from thawing permafrost to the atmosphere, its VOC uptake function may be reduced under drier conditions if all other factors remain unchanged.

The active layer thickness has been observed to increase by different magnitudes in almost all permafrost regions under the context of global warming, such as in Greenland, Qinghai-Tibet Plateau, Siberia, and Alaska[12,45,46]. Hence, the role of active layer soils in mediating the net exchange rates of VOCs between thawing permafrost and the atmosphere is increasingly relevant and warrants further exploration[47]. Future research should focus on disentangling the bidirectional processes rather than solely examining net fluxes, unraveling the contributions of biotic and abiotic sink mechanisms, and identifying the key microbes and metabolism processes involved. Specifically, incorporating the uptake kinetics of VOCs, as informed by the uptake coefficients calculated in this study, into modeling frameworks is strongly recommended. For instance, the potential uptake rates of different VOCs can be approximated by multiplying the coefficients with their concentrations, which could be obtained from in situ observational studies or modeling predictions, in either the ambient air or soil pores. These efforts aim to improve the accuracy of net land-atmosphere VOC exchange rate estimates under future climate conditions and advance our understanding of their associated environmental impacts.

## Methodology
### Soil sample collection
The active layer soil samples were collected from two locations in western Greenland: Disko Island and Kangerlussuaq.

At Disko Island (Blæsedalen, 69.28°N, 53.48°W), the region experiences an Arctic maritime climate, with an annual average air temperature of -3.0 ± 1.8 °C and precipitation of 418 ± 131 mm[48]. The primary ecosystem within this valley is characterized by mesic tundra heath, predominantly featuring both evergreen and deciduous dwarf shrubs, including *Betula nana* L., *Empetrum nigrum* ssp. *hermaphroditum* Hagerup, *Cassiope*

*tetragona* (L.) D. Don, *Salix glauca* L., and *Vaccinium uliginosum* L. These are interspersed with mosses such as *Tomentypnum nitens* (Hedw.) Loeske, *Racomitrium lanuginosum* (Hedw.) Brid., *Sphagnum* spp., and various lichens[49]. The soil was sampled from three depths (10 cm, 20 cm, and 40 cm) at six independent soil pits within an area of 15 × 15 meters, in connection with the establishment of a winter warming experiment[50].

At Kangerlussuaq, soil samples were collected from three sites according to their relative proximity to the continental glacier: glacier site (67.12°N, 50.16°W, closest to the continental glacier), intermediate site (67.06°N, 50.46°W, intermediate), and coast site (67.04°N, 50.55°W, furthest away from the glacier, near coast of Kangerlussuaq Fjord). Within each location, triplicate soil samples ($n = 3$) were collected at three different depths (10, 20, and 40 cm) from soil pits using stainless-steel density ring of approximately 100 cm³. Soil samples were transported back to Copenhagen in plastic bags at temperatures ~5–7 °C and were subsequently stored at −20 °C until the start of the experiments (within three months).

In the laboratory, stones, roots, or visible litter were removed from the soils, which were then homogenized and sieved through a 5 mm mesh. By this, we aimed to exclude the potential interferences originating from the heterogeneity of soils, which allowed us to investigate the processes of VOC production and degradation under a controlled experimental setup.

### Experimental setup for VOC exchange measurements
Soil samples were incubated in pre-conditioned glass jars, and VOC fluxes were measured using a dynamic flow-through method (Supplementary Fig. S2). In practice, soil samples under field-moist conditions (equivalent to 50 g in d.w.) were placed in the 370 ml glass jars with lids fitted with valves for two Teflon lines (inflow and outflow, ¼ inch O.D.). During the VOC measurement period, clean inflow gas, synthetic air containing 20% $O_2$ and 80% $N_2$, was constantly purging the jar at a flow rate of 300 ml min⁻¹ regulated by a gas flow controller in a liquid calibration unit (LCU-a, Ionicon Analytik, Innsbruck, Austria). The outflow air from the sample jars was directed to a high-resolution PTR-ToF-MS instrument (TOF−1000 *ultra*, Ionicon Analytik, Innsbruck, Austria) for real-time continuous measurements of the air.

To streamline and automate the handling of multiple samples, we arranged eight sample jars and one empty blank jar in a parallel configuration (Supplementary Fig. S2). The inflow tube was split and channeled into the nine jars, with the outflow tube from these jars regulated by several PTFE solenoid valves (Cole-Parmer, Cambridgeshire, U.K.). This setup permitted the air to pass through one jar at a time and allowed the analysis of the nine jars in series in a repeated automation manner.

To evaluate the uptake rates of VOCs by active layer soils, a water solution of 12 target VOCs (methanol, acetaldehyde, acetone, isoprene, 2-butanone, 2-methylfuran, toluene, furfural, *cis*-3-hexen-1-ol, benzyl alcohol, terpinolene, and linalool) was vaporized and introduced into the inflow gas stream, creating a headspace of VOCs with certain mixing ratios or potential microbial assimilation. These VOCs were selected based on their water solubility and known relevance to the soil environment[42,51–54]. The vaporization of the VOC solution was regulated using a liquid flow controller within the LCU-a, enabling precise control over the VOC concentrations introduced into the inflow gas stream and jar headspace (Table 3). The resulting concentrations were designed to be comparable to those found in ambient atmospheric conditions of certain environments with significant known sources[55–58].

Automation of both the LCU-a gas and liquid flow rates, and switching of the PTFE solenoid valves were controlled and achieved via the PTR-ToF-MS software (IoniTOF 4.0, Ionicon Analytik, Innsbruck, Austria).

### Experimental design
Different experiments, as followed, were carried out to measure net VOC exchange blends, and the uptake capacity of the soils, and to explore their relationships with soil depth, age and water content to test the hypotheses.

Net exchange profile experiment: soil samples collected from Disko Island ($n = 6$ at each of 10, 20, and 40 cm depths) and Kangerlussuaq (at the

same depths, across three different locations: glacier, intermediate, and coast, with n = 3 for each depth-proximity combination) underwent initial incubation without any VOC introduction into the inflow stream. This setup was designed and conducted to explore the VOC exchange capacity and profiles for all detected masses.

VOC uptake experiment: replicate experiments were conducted under a specific level of VOCs introduced into the inflow stream to assess VOC uptake rates by these active layer soils. For the Disko Island samples, the effect of soil depth on the VOC absorption was evaluated (n = 6 for each depth of 10 cm, 20 cm, and 40 cm,). For the Kangerlussuaq soil samples, the impact of both soil depth (10 cm, 20 cm, 40 cm) and their proximity to the continental glacier (glacier, intermediate, coast) on their VOC uptake capabilities was investigated (n = 3 for each depth-proximity combination).

Water addition experiment: 5 ml of double distilled water (ddH$_2$O) was added to each of the soil samples from both Disko Island and Kangerlussuaq to enhance their water content. These moistened soil samples were then incubated under the same conditions as the field-moist samples, i.e., with a specific level of VOCs introduced into the inflow stream to assess VOC uptake rates. The results were then compared to that of the field-moist samples to evaluate the impact of SWC on VOC uptake.

All three experiments were carried out at 7 °C in a climate chamber (Medilow M 260 L, J.P. SELECTA, Barcelona, Spain) under darkness to reflect the approximate average soil temperature of their origins during the field season. Blank experiments were conducted with blank jars without soil samples in them, while for the water addition experiments, the blank jars were supplemented with 5 ml of ddH$_2$O, as the sample jars. The materials used have been tested to be suitable for VOC measurements[19].

## PTR-ToF-MS measurements

The PTR-ToF-MS required a sample flow of 100 ml min$^{-1}$, and the excess outflowing air (200 ml min$^{-1}$) was released through a vent line. In the drift tube of the PTR-ToF-MS, the pressure was set to 2.30 mbar, the temperature to 60 °C, and the voltage to 500 V, which generated an E/N ratio of 108 Td (1 Td = 10$^{-17}$ V cm$^{-2}$). The ToF acquisition was set to a 5-sec resolution in the mass-to-charge (m/z) range of 30–257 unified atomic mass units (Daltons). A permeation tube containing 1,3-diiodobenzene added a constant signal in the mass spectrum at m/z 203.943 that was used to calibrate the mass scale. The PTR-ToF-MS was calibrated regularly using a standard gas, which was prepared from the aforementioned liquid mixture of the 12 target VOCs, vaporized and diluted in nitrogen gas (>99.999%, Air Liquide Danmark A/S) using the LCU-a. The raw data acquired from the PTR-ToF-MS were processed using the PTRwid software tool[59], which identified the mass peaks in the measured spectra, calibrated the mass scale, and calculated the mixing ratios of the identified masses from the count rates. Afterward, any interfering substances from the mass list were eliminated, such as ion source contaminants (e.g., O$_2$$^+$, NO$_2$$^+$, HCO$_2$$^+$) and known interferences (e.g., hydrate clusters, fragments of 1,3-diiodobenzene). In total, 273 protonated masses were detected during the measurements. Among them, the 12 target compounds in the gas standard were used to directly calibrate the mixing ratios of the PTRwid output, while the remaining protonated masses did not undergo external calibration.

## Flux calculation

We used the convention that positive fluxes represented net VOC emissions from the soils, while negative fluxes represented net VOC uptake from air to soil. VOC fluxes (nmol g dw$^{-1}$ hr$^{-1}$) were calculated based on the equation below:

$$F_{VOC} = \frac{Q \cdot \left( C_{sample} - C_{blank} \right)}{m} \quad (1)$$

Where, Q is the flow rate through the jar, converted to the unit of mol hr$^{-1}$; m is the dry mass of the soil sample (g dw); $C_{sample}$ and $C_{blank}$ are VOC concentrations in the outflow of the soil and blank jars, respectively, and they were determined as the average value of the quasi-stabilized PTR-ToF-

MS reading during each measurement (after ~15–20 min of purging, Supplementary Fig. S3), in the unit of parts per billion by volume (ppbv).

## Uptake coefficient calculation

Assuming that VOC uptake by soil follows first-order kinetics[60,61], i.e., the uptake rate is reversely proportionally with the availability of VOC substrate within a specific concentration range, the uptake coefficients for the different VOCs were calculated.

$$F_{VOC} = -k \left[ C_{VOC} \right] \quad (2)$$

These coefficients (K) were derived as the ratio of the observed uptake rate ($F_{VOC}$) to the VOC concentration in the headspace ($C_{VOC}$, as shown in Table 3, unit: mol g dw$^{-1}$ hr$^{-1}$). A higher uptake coefficient indicates faster VOC uptake by the soil medium. The primary benefit of using uptake coefficients is that they normalize the uptake rates to VOC concentrations, enabling meaningful comparisons of soil uptake across compounds with varying concentrations. Additionally, these coefficients provide potentially valuable parameters for the biogeochemistry modeling community to incorporate soil VOC exchange processes into quantitative kinetic models.

## Analysis of soil properties

DOC, TDN, nitrate (NO$_3$$^-$), and phosphate (PO$_4$$^{3-}$) in the soil samples were analyzed in water extracts after the following procedure: 10 g soil samples were shaken in 50 ml deionized water for 1 hour, and the soil slurry was filtered through Whatman GF-D glass microfiber filters (Whatman Ltd., Maidstone, UK). The soil extract was then measured for DOC and TDN with a TOC-L total organic carbon analyzer (Shimadzu, Kyoto, Japan), and for NO$_3$$^-$ and PO$_4$$^{3-}$, using an FIA STAR 5000 flow injection analyzer (FOSS Tecator, Höganäs, Sweden).

Microbial biomass C (Cmic), N (Nmic), and P (Pmic) were estimated by the chloroform-fumigation-extraction method[62,63]. Cmic, Nmic, and Pmic were calculated as the differences in DOC, TDN, and PO$_4$$^{3-}$, between fumigated and non-fumigated extracts, respectively. To release these elements from the microbes, soil samples were fumigated for 24 h followed by extraction and analysis, as described above. A conversion factor (kEC) of 0.45 was used to compensate for incomplete extractability for C, whilst a kEC of 0.4 was used for N and P[62,63].

Gravimetric SWC was determined by oven-drying the soils to a constant weight at 70 °C and SOM content was estimated by loss on ignition at 550 °C for 6 hours. Total C and N concentrations were determined in the dried and ground soil with a EuroEA3000DF elemental analyzer (Eurovector, Pavia, Italy).

## Statistical analysis

All data, including flux values and soil property data, was log-transformed in order to obtain homogeneous error variances before conducting subsequent statistical analyses, including MANOVA and ANOVA models, which were best suited for our experimental designs rather than rank-based nonparametric tests.

For the experiment that assessed how intrinsic VOC emissions and composition profiles differed between the two locations and across different depth/proximity to glacier, PCA was performed with the log-transformed and unit variance-scaled data of the non-calibrated fluxes of all protonated masses using SIMCA (version 17.0, Umetrics, Umeå, Sweden). The differences in PC scores between the two locations and different depths/proximities to the glacier were evaluated with ANOVA.

For the VOC uptake experiment, the calibrated uptake rates of the 12 target VOCs were used, and the data from Disko Island and Kangerlussuaq were processed separately due to their different experimental designs. To explore variations in soil uptake rates across different soil depths and proximities to the glacier, we employed a MANOVA model, which considered the depth and proximity to the glacier, as well as their interactions, as the main factors (note: only the depth effect was tested for Disko Island data), and the uptake rates of the 12 VOCs as dependent variables. For

factors identified as significant in the MANOVA model, we further examined the individual responses of the 12 VOCs to the factor using ANOVA. In instances where soil depth and glacier proximity significantly influenced a specific VOC's uptake rates, Tukey's HSD post hoc test was applied to discern variations in VOC uptake rates across different depths and proximities Pearson correlation analysis was conducted to assess the potential relationships between uptake rates of the VOCs and various soil properties.

For the water addition experiment, the same statistical analyses that were used in the VOC uptake experiment were used to investigate the impact of water addition. Differences in soil properties between the two locations and the differences across soil depth and proximity to glaciers within each site were explored separately by MANOVA and ANOVA.

All statistical analyses, except for the PCA, were performed using IBM SPSS Statistics (version 29.0.2.0), with a threshold of $P < 0.05$ considered significant.

## Reporting summary
Further information on research design is available in the Nature Portfolio Reporting Summary linked to this article.

## Data availability
All data that support the findings of this study is presented in the manuscript, the supplementary information and archived dataset (doi: 10.5281/zenodo.14185189).

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

## Acknowledgements
This study was supported by Horizon Europe Marie Skłodowska-Curie Action (Postdoctoral Fellowship No. 101061660 to Y.J.), Danish National Research Foundation Center of Excellence (grant No. DNRF168 to R.R., & grant No. DNRF1000 to B.E.), European Research Council (consolidator grant No. 771012 to R.R.), and Danish Ministry for Higher Education and Science (Elite Research Prize No. 9095-00004 to R.R.). The authors would like to thank Gosha Sylvester for the assistance in soil property measurements. No permission was required to access our field sites in Greenland.

## Author contributions
Y.J., R.R., and M.K. conceived the study. M.K., B.E., and R.R. collected the samples. M.K. and C.D.-M. conducted the experiments. Y.J. led the formal analysis and visualization, with contributions from C.D.-M. and M.K. Y.J. prepared the manuscript, which was then reviewed, edited, and approved by all authors.

## Competing interests
The authors declare no competing interests.
