## [Transparent Peer Review file · Communications Earth & Environment]

The active layer soils of Greenlandic permafrost areas can function as important sinks for volatile organic compounds

Corresponding Author: Professor Riikka Rinnan

Version 0:

Decision Letter:

Dear Professor Rinnan,

Your manuscript titled "The active layer soils of Greenlandic permafrost areas can function as important sinks for volatile organic compounds" has now been seen by 2 reviewers, whose comments are appended below. You will see that they find your work of potential interest. However, they have raised quite substantial concerns that must be addressed. In light of these comments, we cannot accept the manuscript for publication in its current form, but would be interested in considering a revised version that fully addresses these concerns.

In particular we require that you:

- demonstrate that Greenlandic permafrost soil can function as a sink for VOCs
- provide clear evidence that microbial activity (rather than other drivers) is creating the sink in VOCs in Greenlandic permafrost soil.

We hope you will find the reviewers' comments useful as you decide how to proceed. Should additional work allow you to address these criticisms, we would be happy to look at a substantially revised manuscript. If you choose to take up this option, please either highlight all changes in the manuscript text file, or provide a list of the changes to the manuscript with your responses to the reviewers.

When resubmitting, please provide a point-by-point response to the reviewers' comments. Please submit your responses as a separate file, distinct from your cover letter where you can add responses to the Editors' comments that you do not want to be made available to the reviewers. Word files are preferred. We recommend that any figures, tables or graphs that are included in the response to reviewers are also included in the main article or Supplementary Information.

If the revision process takes significantly longer than three months, we will be happy to reconsider your paper at a later date, as long as nothing similar has been accepted for publication at Communications Earth & Environment or published elsewhere in the meantime.

Please use the following link to submit your revised manuscript, point-by-point response to the reviewers' comments with a list of your changes to the manuscript text (which should be in a separate document to any cover letter), a tracked-changes version of the manuscript (as a PDF file) and any completed checklist:

Link Redacted

Please do not hesitate to contact us if you have any questions or would like to discuss the required revisions further. Thank you for the opportunity to review your work.

Best regards,

Alice Drinkwater, PhD
Associate Editor
Communications Earth & Environment
@CommsEarth

EDITORIAL POLICIES AND FORMAT

If you decide to resubmit your paper, please ensure that your manuscript complies with our editorial policies and complete and upload the checklist below as a Related Manuscript file type with the revised article:

Editorial Policy Policy requirements
(Download the link to your computer as a PDF.)

- Behavioural and social science
- Ecological, evolutionary & environmental sciences
- Life sciences

<https://www.nature.com/documents/nr-reporting-summary.zip>

For your information, you can find some guidance regarding format requirements summarized on the following checklist: (<https://www.nature.com/documents/commsj-phys-style-formatting-checklist-article.pdf>) and formatting guide (<https://www.nature.com/documents/commsj-phys-style-formatting-guide-accept.pdf>).

REVIEWER COMMENTS:

Reviewer #2 (Remarks to the Author):

This article describes the biogeochemical cycling of VOCs from soils collected from contrasting locations in Greenland at different depths. The soil samples were subjected to measure net VOC emissions under laboratory conditions. Later, soil samples were exposed to a VOC mixture, and their consumption rate was determined. In addition, the role of soil water content was tried to evaluate on the exchange rate of VOCs. The article is well-written and has a good text flow, but it lacks novelty and depth of study. No list of tentatively identified VOCs emitted from soils at different depths is provided. Figures lack statistical information, hard to see significant differences. The soil physicochemical properties were determined but no correlation analysis was done with the VOC profiles. Authors always argue that microbial activity is the main sink factor other than binding with organic matter and clay particles, but no evidence is provided in this study. The overall results are not novel and have already been reported in several studies in one way or another. For the current study, we can only see there is a difference in VOC emission (data missing) and sink capacity of soils collected from different locations at different depths, but no further evidence is provided that why both locations differed in their capacity.

Reviewer #3 (Remarks to the Author):

Review Comments:

The main claim of the manuscript is that the active layer of Arctic soil may act as a significant sink for VOCs emitted by melting permafrost. While there have been previous studies on the Arctic active layer acting as a sink for certain VOCs (e.g., Kramshøj et al., 2018), this study provides deeper insights into the phenomenon and demonstrates that this is true for soils with different physicochemical properties. The authors could further clarify the novelty of the manuscript. The manuscript is certainly of interest to the VOC and Arctic research communities. With its extensive set of Arctic soil samples, state-of-the-art analyses, and thorough data interpretation, the work is convincing. However, additional evidence could be provided to strengthen the conclusions, as detailed below.

The authors claim that the sink is important. However, based on the data presented in the manuscript, it is not possible to estimate its significance. They should provide an estimate of how important this sink is, for example, compared to potential permafrost emissions, emissions from ground vegetation, or compared to atmospheric chemical sinks. Currently, it is unclear what fraction of the VOCs injected into the soil chambers was taken up. Is this fraction the uptake coefficient? The concept of the uptake coefficient was very unclear, and the supplementary table, which was supposed to clarify it, was missing.

Detailed Comments:

- It is well known that many VOCs are easily lost on different kinds of surfaces, and the surface activity of these compounds varies, which could also explain the variation in the uptake coefficient. Could it be possible to use some artificial soil for comparison to verify that the VOCs were lost due to microbial activity and not just surface activity?
- Did you investigate whether VOCs were re-emitted after stopping the VOC injection? Could this provide confirmation that they were consumed by microbes rather than just lost on surfaces?
- The sink was enhanced when the soil was moist. Many of these compounds are polar/water-soluble. Please clarify why you expect that this was due to enhanced uptake rather than just due to losses on moist surfaces.
- Section 3.2: You mention that terpinolene was the main emitted compound. How do you know it was terpinolene and not some other monoterpene if you were measuring with the PTR-TOF? Could you add emission rates for the main emitted compounds and for the compounds you used for the uptake rate studies? It would enable the comparison to the uptake rates.
- Were there losses in the blank chambers? How significant were these compared to the experimental chambers? Could you include some data on the blank chambers?
- Section 3.5: Please explain the uptake coefficient more clearly and provide the equation used for its calculation. What is meant by "headspace"? Does it refer to the concentration in the VOC-rich air flushed into the chambers shown in Table 1? If the uptake coefficient represents the fraction of VOCs taken up by the soil, it would be very interesting to see more detailed data on this, especially since, as you mention, uptake rates depend on the incoming concentrations of VOCs.
- Line 410: Glass is not a particularly surface-active material and is often recommended for VOC studies as a surface material. Perhaps using another material more like soil would provide a better understanding.

Reference: Kramshøj, M., Albers, C. N., Holst, T., Holzinger, R., Elberling, B., & Rinnan, R. (2018). Biogenic volatile release from permafrost thaw is determined by the soil microbial sink. *Nature Communications*, 9(1), 3412.
<https://doi.org/10.1038/s41467-018-05824-y>

Communications Earth & Environment is committed to improving transparency in authorship. As part of our efforts in this direction, we are now requesting that all authors identified as 'corresponding author' create and link their Open Researcher and Contributor Identifier (ORCID) with their account on the Manuscript Tracking System prior to acceptance. ORCID helps the scientific community achieve unambiguous attribution of all scholarly contributions. You can create and link your ORCID from the home page of the Manuscript Tracking System by clicking on 'Modify my Springer Nature account' and following the instructions in the link below. Please also inform all co-authors that they can add their ORCIDs to their accounts and that they must do so prior to acceptance.

Version 1:

Decision Letter:

Dear Professor Rinnan,

Your manuscript titled "The active layer soils of Greenlandic permafrost areas can function as important sinks for volatile organic compounds" has now been seen by our reviewers, whose comments appear below. In light of their advice we are delighted to say that we are happy, in principle, to publish a suitably revised version in *Communications Earth & Environment*.

We therefore invite you to revise your paper one last time to address the remaining concerns of our reviewers. At the same time we ask that you edit your manuscript to comply with our format requirements and to maximise the accessibility and therefore the impact of your work.

EDITORIAL REQUESTS:

*****Please take care to match our formatting and policy requirements. We will check revised manuscript and return manuscripts that do not comply. Such requests will lead to delays. *****

SUBMISSION INFORMATION:

OPEN ACCESS:

Communications Earth & Environment is a fully open access journal. Articles are made freely accessible on publication. For further information about article processing charges, open access funding, and advice and support from Nature Research, please visit <https://www.nature.com/commsenv/open-access>

Link Redacted

Best regards,

Alice Drinkwater, PhD
Associate Editor
Communications Earth & Environment
@CommsEarth

REVIEWERS' COMMENTS:

Reviewer #2 (Remarks to the Author):

The authors provide convincing answers by providing more data and rewriting the discussion of the manuscript. It can not be called a comprehensive study, including some questions raised by other reviewers. However, it provides enough information to the conclusion of the study. I have only some minor comments.

- It is better to perform PERMANOVA to see differences in VOC profiles.
- Provide actual P values in figures and text.
- In Table 1 and Table 3: write VOC names with first capital letters.
- Fig. 4 and Fig. 5 legends need more information, overall VOC uptake was different but did individual VOC uptake not differ?

Reviewer #3 (Remarks to the Author):

Thank you for your thorough response to my previous comments on your manuscript. I appreciate the effort, but I still have a

few additional suggestions and clarifications:

1. I feel the importance of this process is still not clearly conveyed. There might have been some misunderstanding regarding my question about the fraction of VOCs injected into the soil chambers that were taken up by the soil. If I understood correctly, you injected a calibration gas into the chamber with known concentrations (C_1) as listed in Table 1, then measured the outflow concentrations (C_2). From this, it should be straightforward to calculate and present the fraction taken up by the soil using the formula: $(C_1 - C_2)/C_1$. This would provide a clearer indication of the magnitude of uptake. Relying solely on uptake rates and uptake coefficients makes it difficult to assess how significant the uptake is. For instance, is it less than 1% (well within the uncertainty of such measurements) or closer to 100%? Based on the figure in your rebuttal letter (Selected VOC concentration change over time for the nine jars: 8 soil jars averaged + 1 blank jar), it seems that 50–75% of the VOCs were taken up, indicating strong uptake at these concentration levels.
2. Supplementary Table S1: The unit of emission rates is missing in this table. Please include it for clarity.
3. Line 487–490: You mention toluene, but toluene is only very slightly soluble and is usually considered non-soluble. On the other hand, acetaldehyde is miscible yet shows lower uptake rates with water addition. Perhaps you could revise your argument to focus on acetaldehyde instead of toluene, or simply remove toluene from this section.
4. Section 3.5: VOC uptake coefficients: What is the unit for the uptake coefficients? Please include the unit in Table 3.
5. On line 496: You mention that uptake coefficients can be used for modeling. Could you elaborate on how this would be done? For instance, would it require knowledge of permafrost emissions without an active soil layer? Is such information already available? Would you also need to model air concentrations within the soil? A brief explanation would improve clarity here.

Point-by-point responses to the comments from the reviewers

We would like to express our sincere gratitude to the editor and the reviewers for their valuable time and thoughtful evaluation of our manuscript. We have carefully considered all the comments and concerns raised during the review process and have made significant revisions in response. We consider these constructive suggestions have helped us to greatly improve the clarity and overall quality of the manuscript. In this response letter, we begin with an overview of the key revisions made, and then point-by-point responses to each of the comments from the reviewers.

We acknowledge the reviewers' main concerns regarding the depth of the study. Hence, we have incorporated more quantitative calculations and highlighted the new and distinct findings of this work. Below is a summary of the key revisions made to improve the manuscript, with detailed explanations provided in response to each reviewer comment:

- Rewrote the entire discussion section to incorporate the points raised by the reviewers, such as evidence supporting the likely existence of the biotic sinks, the implication of changes in the water content on VOC exchanges.
- Provided a table of the uptake coefficients of individual VOCs, which can potentially be used by modelling communities, and also reflect the potential efficiency of the soil sink for VOCs. Accordingly, we have added a method section describing the calculation and meaning of the uptake coefficients.
- Performed correlation analysis between soil VOC fluxes and physicochemical properties to provide insights into the factors driving VOC uptake.
- Provided a full list of tentatively emitted compounds in the data repository, showed the contributions from the top 10 emitted protonated masses in a supplementary table.
- Updated figures and their captions with statistical information or labels to highlight significant factors or differences between treatments and groups.

*****Reviewer #2*****

This article describes the biogeochemical cycling of VOCs from soils collected from contrasting locations in Greenland at different depths. The soil samples were subjected to measure net VOC emissions under laboratory conditions. Later, soil samples were exposed to a VOC mixture, and their consumption rate was determined. In addition, the role of soil water

content was tried to evaluate on the exchange rate of VOCs. The article is well-written and has a good text flow, but it lacks novelty and depth of study.... *(To be continued)*

Response: We thank the reviewer for the positive summary of our work and the appreciation of the manuscript's structure and flow. We also acknowledge the concerns on the novelty and depth of the study. Please kindly refer to the detailed response to your specific comments below.

Permafrost thaw and the magnitude of the climate change-induced carbon losses from the permafrost-affected soils are timely issues with broad implications. Recent studies have shown that a fraction of this carbon is released as volatile organic compounds (VOCs), reactive gases that can influence the air quality and the Earth's radiative balance through feedbacks related to the formation of tropospheric ozone and secondary organic aerosols, and cloud condensation nuclei. However, whether the VOCs released from permafrost would be emitted to the atmosphere has been an open question. A previous work from our lab (Kramshøj et al., 2018, *Nat. Commun.*) showed that the active layer on top of the permafrost lay can uptake several VOCs, especially the low molecular mass (e.g., ethanol, methanol, acetone, formaldehyde), released from the thawing permafrost. Building on this foundational work, our current study expands the scope to test the potential uptake of a broader range of VOC species, including larger compounds with significant environmental implications (e.g., isoprene, linalool, toluene, etc.), as well as across multiple sites and soil depths of the Greenlandic active layer to explore its spatial extent. Our results confirm that the soils in the active layer act as a strong sink not only for low molecular mass VOCs but also for larger compounds, such as isoprene and terpenoids. The uptake processes were observed across multiple sites and depths of the active layer in Greenland, which demonstrated the active layer's crucial role in regulating the net exchange of these gases at the soil-atmosphere interface as permafrost thaws.

...(Continuing) No list of tentatively identified VOCs emitted from soils at different depths is provided. ... *(To be continued)*

Response: We have provided a list of tentatively identified protonated masses emitted by the active layer soils across different sites and depths. The top 10 emitted masses and their emission rates are provided in Supplementary Table S1). A full mass list of the tentative compounds emitted by these soils, along with their emission rates, is

available in the supplementary dataset (<https://doi.org/10.5281/zenodo.14185189>, the link will be available to the public upon the publication)

...(Continuing) Figures lack statistical information, hard to see significant differences. ... (To be continued)

Response: In our original submission, the statistical analysis results (including MANOVA, ANOVA and post-hoc tests) were provided in the accompanying result text, where the figures were interpreted. In light of the reviewer's comment, we have updated figures and their captions with statistical information to highlight the significant factors or differences between treatments and groups. Please kindly refer to the updated Figs. 2-5 to see the changes.

...(Continuing) The soil physicochemical properties were determined but no correlation analysis was done with the VOC profiles. ... (To be continued)

Response: In light of the reviewer's comment, we have conducted Pearson correlation analysis to evaluate the potential relationships between soil physicochemical properties and the uptake rates of the 12 targeted VOCs (Figure attached below, also included in the revision). This analysis provides information on the soil characteristics (presented in Table 2) are associated with VOC uptake rates. Our results show that soil moisture, SOM, Total C and N, and particularly microbial biomass carbon, exhibit significant correlations with the uptake rates of most of the 12 VOC compounds, highlighting the importance of soil type for VOC uptake and a potential link between VOC uptake and soil microbial activity. The corresponding figure (**Supplementary Fig. 2**) and relevant discussion (**Lines 26-27, 463-465**) has been included in the revised manuscript.

Supplementary Fig. 2 Heatmap showing correlation coefficients between the uptake rates of the VOCs and the soil properties. Heatmap was colored to indicate the direction of the correlation, with red representing positive correlations and blue representing negative correlations; number in each grid cell shows the Pearson’s correlation coefficient, labeled with * for significance at $P < 0.05$ and ** for significance at $P < 0.01$; SWC, gravimetric soil water content; SOM, soil organic matter; DOC, dissolved organic carbon; TDN, total dissolved nitrogen; Cmic, microbial biomass carbon; Nmic, microbial biomass nitrogen; Pmic, microbial biomass phosphorus.

...(Continuing) Authors always argue that microbial activity is the main sink factor other than binding with organic matter and clay particles, but no evidence is provided in this study. ... (To be continued)

Response: We thank the reviewer for raising this important point. The claim of microbial uptake of VOCs in these soils is supported by multiple lines of evidence, including those from our previous publications.

New evidence from this study: We calculated uptake coefficients for different VOCs (the same coefficient is called deposition velocity in the field of physics). The variation

in these coefficients suggests that the uptake is not solely physical, as we would expect more uniform values if physical processes were dominant. The observed variation in uptake coefficients for different compounds suggests that microbes may exhibit “dietary preferences”. Furthermore, in the additional analyses performed for the revision, we also tested the correlations between soil properties and VOC uptake rates as suggested by the reviewer, and found a significant positive relationship between microbial biomass carbon and VOC uptake. Collectively, this set of evidence suggests that VOC uptake in these soils is likely dominated by microbial activity, although contributions from physical processes, such as adsorption to organic matter or clay particles, cannot be fully excluded.

Evidence from the previous publications of our group: Isotope-labelling experiments have demonstrated rapid conversion of ^{14}C -VOCs into $^{14}\text{CO}_2$ (mineralization) by soils, a process that has taken place for all tested VOCs from simple to complex molecules (Kramshøj et al., 2018; Albers et al., 2018). These experiments were conducted also after autoclaving the soil, and no mineralization activity was observed at all. Additionally, our temperature-dependent incubation studies (the identical setup as this study) showed that VOC uptake rates increased with temperature, further indicating that the process is not primarily driven by physical adsorption or dissolution (Jiao et al., 2023). If physical processes would dominate, uptake rate should in fact decrease with increasing temperature, not increase as we observed.

In response to your comments, we have rewritten the discussion section to consolidate and summarize all of the evidence gathered from our experiments (**Lines 401-422, Lines 459-474**). Additionally, we have adjusted the strength of our claims, acknowledging that while our findings strongly support microbial VOC uptake, we cannot entirely rule out the possibility of abiotic degradation and physical adsorption (**Lines 449-458**), as suggested by the reviewer.

...(Continuing) The overall results are not novel and have already been reported in several studies in one way or another. For the current study, we can only see there is a difference in VOC emission (data missing) and sink capacity of soils collected from different locations at different depths, but no further evidence is provided that why both locations differed in their capacity.

Response: Our study builds on previous research, but has several novel elements. (1) we tested a broad range of VOCs, including those with higher reactivity, such as isoprene, linalool, and toluene, to better understand the soil's role in VOC uptake, particularly for compounds with greater atmospheric chemistry implications. We focused on permafrost areas in order to assess the sink capacity of the active layer soils that are on top of the thawing permafrost layers, which are a considerable VOC source (Li et al., 2020; Jiao et al., 2023). (2) To our knowledge, this is the first study to examine how VOC uptake differs across the soil depth gradient and across the coast-glacier gradient with different soil ages. This is important to understand the spatial variation and to be able to model the processes in the soil profile. (3) Different units and methods have been used in the few studies that assess soil VOC uptake complicating comparisons. Here, we have provided a more quantitative assessment of the sink process by calculating the uptake coefficients for 12 target VOC species, which can be utilized by the modeling community to incorporate soil VOC uptake into their frameworks. In light of the reviewer's comment, we have restructured the manuscript to sharpen these messages. In terms of the comments on VOC emission data, please kindly refer to our response to one of your comments above.

*****Reviewer #3*****

The main claim of the manuscript is that the active layer of Arctic soil may act as a significant sink for VOCs emitted by melting permafrost. While there have been previous studies on the Arctic active layer acting as a sink for certain VOCs (e.g., Kramshøj et al., 2018), this study provides deeper insights into the phenomenon and demonstrates that this is true for soils with different physicochemical properties. The authors could further clarify the novelty of the manuscript. The manuscript is certainly of interest to the VOC and Arctic research communities. With its extensive set of Arctic soil samples, state-of-the-art analyses, and thorough data interpretation, the work is convincing. However, additional evidence could be provided to strengthen the conclusions, as detailed below.

Response: We thank the reviewer for the generally positive appraisal of our work. Your comments and suggestions have been addressed in a point-by-point manner as detailed below.

The authors claim that the sink is important. However, based on the data presented in the manuscript, it is not possible to estimate its significance. They should provide an estimate of

how important this sink is, for example, compared to potential permafrost emissions, emissions from ground vegetation, or compared to atmospheric chemical sinks. Currently, it is unclear what fraction of the VOCs injected into the soil chambers was taken up. Is this fraction the uptake coefficient? The concept of the uptake coefficient was very unclear, and the supplementary table, which was supposed to clarify it, was missing.

Response: We thank the reviewer for raising this important point. In our original submission, we introduced the uptake coefficient, which is a similar but slightly different parameter to the “uptake fraction”, as suggested by the reviewer. To calculate the uptake coefficients, we assumed that the uptake of VOCs by soil processes (either biotic or abiotic) follow a first-order kinetics. This means that the uptake rate increases with the concentration of VOCs in the headspace (i.e., concentration in the VOC-rich air flushed into the jars, as shown in Table 1) within a certain concentration range. Hence, it was calculated as the ratio of the observed uptake rate to the headspace concentration of the VOC. A higher coefficient indicates faster VOC uptake by the soil medium. The benefit is that, after normalizing the uptake rates to the headspace concentration of the VOC (i.e., uptake coefficient), it becomes easier to compare between-compound differences in soil uptake. Additionally, we provided the uptake coefficients because they can be useful for the modeling community to incorporate the sink kinetic processes in a quantitative manner (**Table 3**). In the revision, we have also added the calculation of uptake coefficients in the method section (**Lines 213-225**).

We agree with the reviewer that knowing the fraction of VOCs injected into the soil chambers that was taken up by the soil would be an important data to report. However, our setup with the dynamic flow-through system is not ideal for assessing that. A static measurement technique might be more suitable, and we have made a note of it, and will include this in our follow-up experiments.

Detailed Comments:

It is well known that many VOCs are easily lost on different kinds of surfaces, and the surface activity of these compounds varies, which could also explain the variation in the uptake coefficient. Could it be possible to use some artificial soil for comparison to verify that the VOCs were lost due to microbial activity and not just surface activity?

Response: We have noticed that the reviewer has raised several relevant points along the same line of concern, and we would like to address them collectively. As mentioned

in our response to a similar concern raised by another reviewer, it is shown that the uptake of VOCs by these soils was predominantly driven by biotic sinks, i.e., microbial uptake, rather than reversible processes, such as physical adsorption or dissolution, as supported by several lines of evidence (e.g., statistically significant correlation with microbial biomass carbon), including our previous publications (Lines 401-422, Lines 459-474).

While we consider microbial uptake is the primary factor driving VOC uptake, we acknowledge that other processes like physical adsorption or dissolution may also contribute (Lines 449-453). We agree with the reviewer that glass beads were not an ideal material for replicating the adsorption properties of soils. However, it was challenging to find alternative materials that could match the soil's adsorption capacity. We previously attempted comparison studies using sterilized soil samples in similar experiments. However, in practice, they exhibited several-fold higher VOC emissions after sterilization by either autoclaving or gamma irradiation. This increase was likely due to the lysis of microbial cells, and degradation of labile soil carbon under heat or irradiation, making sterilized soils unsuitable as non-biotic controls for VOC measurements. We have added a sentence of discussion on the potential future experiments using other materials as alternative soil particles (Lines 415-417).

“However, glass beads may not fully replicate the complex surface properties of soil particles, and future studies could consider using crushed rocks or other materials to better mimic soil surfaces.” (Lines 415-417)

As the reviewer suggested, observing potential VOC re-emission after stopping the injection could be an additional way to test whether the sink is reversible. We have taken note of this for future experimental designs and included it as a suggestion for further studies in the discussion of this manuscript (Lines 453-458).

“It is also worth noting that the uptake incubations in this study were relatively short in duration, making it unclear how much physical adsorption or dissolution may have contributed to the observed uptake. Future studies involving longer incubation periods or continuous monitoring of VOC concentrations after stopping the VOC supply would

be valuable to identify any potential re-emission and better understand the role of physical processes in VOC uptake.” (Lines 453-458).

Did you investigate whether VOCs were re-emitted after stopping the VOC injection? Could this provide confirmation that they were consumed by microbes rather than just lost on surfaces?

Response: Please kindly refer to our response to one of your above comments, and the new discussion quoted below.

“Although the microbial uptake of VOCs in the active layer likely dominates, other processes, such as physical adsorption, dissolution, and chemical degradation, can also contribute. Consequently, the uptake of VOCs by soils is influenced by a combination of factors, including soil physicochemical properties, environmental conditions, and the microbial community and its activity. It is also worth noting that the uptake incubations in this study were relatively short in duration, making it unclear how much physical adsorption or dissolution may have contributed to the observed uptake. Future studies involving longer incubation periods or continuous monitoring of VOC concentrations after stopping the VOC supply would be valuable to identify any potential re-emission and better understand the role of physical processes in VOC uptake.” (Lines 449-458)

The sink was enhanced when the soil was moist. Many of these compounds are polar/water-soluble. Please clarify why you expect that this was due to enhanced uptake rather than just due to losses on moist surfaces.

Response: Soil microbial processes are often inhibited in very dry soils, which could also apply to microbial VOC uptake in our study, as the field-moist soils were relatively dry (e.g., SWC 0.4–11% for the Kangerlussuaq soils). The most direct evidence in our study supporting our argument is probably that the uptake coefficients of VOCs did not consistently become more negative (i.e., indicating higher uptake potentials). For instance, VOCs such as toluene (solubility in water at 20°C: 0.52 g/L), benzyl alcohol (40 g/L), and furfural (83 g/L) showed lower uptake potentials after water addition for the Kangerlussuaq soils, despite being water-soluble (Lines 407-412). This suggested that physical dissolution alone was not the sole driver of the observed VOC uptake in our study. However, we also agree with the reviewer’s suggestion. In the revised manuscript, we have toned down the statements regarding the explanation for enhanced

uptake after moistening and have acknowledged that dissolution could also be an important factor contributing to the observed enhancement (Lines 449-458).

Section 3.2: You mention that terpinolene was the main emitted compound. How do you know it was terpinolene and not some other monoterpene if you were measuring with the PTR-TOF? Could you add emission rates for the main emitted compounds and for the compounds you used for the uptake rate studies? It would enable the comparison to the uptake rates.

Response: We applied a mixture of external gases, including terpinolene as a monoterpene, to conduct calibrations with the PTR-ToF-MS. We agree with the reviewer that the mass emitted could be any monoterpene or other compounds with the same mass as terpinolene. We have removed such statement in the revision. Additionally, we calculated the ratio of emission and uptake rates for the target compounds. None of the 12 compounds targeted in the uptake experiments accounted for more than 10% of the total emission rates. As suggested, the top 10 emitted compounds, along with the full list of emitted compounds and their emission rates, are provided in the Supplementary Table S1 and the archived dataset.

Were there losses in the blank chambers? How significant were these compared to the experimental chambers? Could you include some data on the blank chambers?

Response: The blank jars showed minimal changes in VOC concentrations. Please kindly refer to the following figures, where the blank jars and sample jars are plotted on the same axis for comparison. When calculating the fluxes, the VOC signals from the blank jars were subtracted from those of the sample jars to account for background levels, under the assumption that the blank jars represent the baseline conditions.

Selected VOC concentration change versus time of the 9 jars (8 soil jars averaged + 1 blank jar) during the experiments with and without VOCs amended into the inlet gas

Section 3.5: Please explain the uptake coefficient more clearly and provide the equation used for its calculation. What is meant by "headspace"? Does it refer to the concentration in the VOC-rich air flushed into the chambers shown in Table 1? If the uptake coefficient represents the fraction of VOCs taken up by the soil, it would be very interesting to see more detailed data on this, especially since, as you mention, uptake rates depend on the incoming concentrations of VOCs.

Response: Yes, headspace concentration refers to the concentration in the VOC-rich air flushed into the chambers shown in Table 1. As suggested by the reviewer, we have added the method section on how to calculate the uptake coefficient (quoted below).

“Assuming that VOC uptake by soil follows first-order kinetics, i.e., the uptake rate is reversely proportionally with the availability of VOC substrate within a specific concentration range, the uptake coefficients for the different VOCs were calculated.

$$F_{VOC} = -k[C_{VOC}] \quad (2)$$

These coefficients (K) were derived as the ratio of the observed uptake rate (FVOC) to the VOC concentration in the headspace (CVOC, as shown in Table 1). A higher uptake coefficients indicates faster VOC uptake by the soil medium. The primary benefit of using uptake coefficients is that they normalize the uptake rates to VOC concentrations, enabling meaningful comparisons of soil uptake across compounds with varying concentrations. Additionally, these coefficients provide potentially valuable parameters for the biogeochemistry modeling community to incorporate soil VOC exchange processes into quantitative kinetic models.” (Lines 214-226)

“Specifically, incorporating the uptake kinetics of VOCs, as informed by the uptake coefficients calculated in this study, into modeling frameworks is strongly recommended.” (Lines 495-497)

Line 410: Glass is not a particularly surface-active material and is often recommended for VOC studies as a surface material. Perhaps using another material more like soil would provide a better understanding.

Response: We agree with the reviewer that glass beads were not an ideal material for replicating the adsorption properties of soils. However, it was challenging to find alternative materials that could match the soil's adsorption capacity. We have added a sentence in the discussion (quoted below) suggesting that crushed rock materials may be considered to be used for replicating the surface of soil particles.

“However, glass beads may not fully replicate the complex surface properties of soil particles, and future studies could consider using crushed rocks or other materials to better mimic soil surfaces.” (Lines 415-417)

Reference: Kramshøj, M., Albers, C. N., Holst, T., Holzinger, R., Elberling, B., & Rinnan, R. (2018). Biogenic volatile release from permafrost thaw is determined by the soil microbial sink. *Nature Communications*, 9(1), 3412. <https://doi.org/10.1038/s41467-018-05824-y>

Point-by-point responses to the comments from the reviewers

We would like to express our sincere gratitude to the editor and reviewers for their valuable time and thoughtful evaluation of our manuscript. We are also pleased to hear that our previous revisions have addressed many of the concerns raised. In this round of revisions, we have carefully considered all new comments and incorporated the suggested changes, which we believe have further improved the manuscript and made it suitable for publication. Below, we provide a point-by-point response to the reviewers' comments. Unless otherwise specified, the line numbers (Lines ###) refer to the clean copy of the revised manuscript.

*****Reviewer #2*****

The authors provide convincing answers by providing more data and rewriting the discussion of the manuscript. It can not be called a comprehensive study, including some questions raised by other reviewers. However, it provides enough information to the conclusion of the study. I have only some minor comments.

Response: Thanks for your recognition. The new comments have also been fully considered and incorporated. Please find our point-by-point responses below.

It is better to perform PERMANOVA to see differences in VOC profiles.

Response: We appreciate the reviewer's suggestion to use PERMANOVA for analyzing differences in VOC profiles. While PERMANOVA is a valuable multivariate method, we believe MANOVA is more appropriate for our dataset and research focus. Since our study examines the effects of categorical variables (e.g., soil depth, proximity to glaciers) on overall VOC uptake rates across multiple response variables, MANOVA is well-suited for this purpose. Importantly, our dataset satisfies the assumptions required for MANOVA, including multivariate normality and homogeneity of variances. We acknowledge that PERMANOVA is particularly useful for datasets that do not meet parametric assumptions or involve distance-based analyses, such as community composition studies. However, in our case, using MANOVA enables us to leverage the parametric nature of our dataset and maintain the full structure of the data, which aligns better with the goals of this study.

We are grateful for the reviewer's insight and will consider using PERMANOVA in future studies where its strengths may be more applicable.

Provide actual P values in figures and text.

Response: Thank you for the suggestion. In the revised manuscript, we have updated the manuscript to include actual P values wherever appropriate to enhance clarity and transparency.

In Table 1 and Table 3: write VOC names with first capital letters.

Response: Corrected as suggested.

Fig. 4 and Fig. 5 legends need more information, overall VOC uptake was different but did individual VOC uptake not differ?

Response: We have updated the legends of Fig. 4 and Fig. 5 to include information on the differences between field-moist soils and moistened soils. Take Kangerlussuaq soils as an example (Fig. 5), moistened soils also showed increased uptake rates compared to their field-moist counterparts, with the exception of acetaldehyde ($P < 0.001$, ANOVA), furfural ($P < 0.001$, ANOVA), and toluene ($P = 0.006$, ANOVA), which displayed contrary trends. Meanwhile, 2-butanone ($P = 0.741$, ANOVA), isoprene ($P = 0.333$, ANOVA), and benzyl alcohol ($P = 0.741$, ANOVA) did not exhibit significant differences in uptake rates between the two moisture levels. This information has now been incorporated into the figure legends.

*****Reviewer #3*****

Thank you for your thorough response to my previous comments on your manuscript. I appreciate the effort, but I still have a few additional suggestions and clarifications:

Response: We thank the reviewer for the feedback and positive evaluation of our study. Please find our point-by-point responses to the additional suggestions and clarifications below.

1. I feel the importance of this process is still not clearly conveyed. There might have been some misunderstanding regarding my question about the fraction of VOCs injected into the soil chambers that were taken up by the soil. If I understood correctly, you injected a calibration gas into the chamber with known concentrations (C1) as listed in Table 1, then measured the

outflow concentrations (C2). From this, it should be straightforward to calculate and present the fraction taken up by the soil using the formula: $(C1-C2)/C1$. This would provide a clearer indication of the magnitude of uptake. Relying solely on uptake rates and uptake coefficients makes it difficult to assess how significant the uptake is. For instance, is it less than 1% (well within the uncertainty of such measurements) or closer to 100%? Based on the figure in your rebuttal letter (Selected VOC concentration change over time for the nine jars: 8 soil jars averaged + 1 blank jar), it seems that 50–75% of the VOCs were taken up, indicating strong uptake at these concentration levels.

Response: Thank you for your comment. In this study, we used a dynamic method, where VOCs were continuously purged through the soil chambers, and the outflow concentrations were measured in real-time. Due to this dynamic setup, the actual fraction of VOCs taken up by the soils may not be straightforward to calculate, as the concentrations were not stabilized until approximately 20–40 minutes after purging began. Additionally, long-term observations were not conducted in this study, which further limits our ability to determine the steady-state fraction of uptake.

Instead, to provide a clearer picture of the uptake dynamics, we included a plot showing the outflowing concentration change over time in the **supplementary Fig. S3**. This figure allows readers to visualize the time-dependent behavior of VOC uptake and how the outflow concentrations evolve in comparison to the blank measurements, which may help in interpreting the uptake process.

2. Supplementary Table S1: The unit of emission rates is missing in this table. Please include it for clarity.

Response: Thanks for the suggestions. The unit of emission rates ($\text{nmol g dw}^{-1} \text{ hr}^{-1}$) is now added in the table title.

3. Line 487–490: You mention toluene, but toluene is only very slightly soluble and is usually considered non-soluble. On the other hand, acetaldehyde is miscible yet shows lower uptake rates with water addition. Perhaps you could revise your argument to focus on acetaldehyde instead of toluene, or simply remove toluene from this section.

Response: Thank you for pointing this out. We agree that toluene is very slightly soluble (solubility in water at 20°C: 0.52 g L^{-1}). In light of your suggestion, we removed

it in this section, and added acetaldehyde (miscible, $\sim 1000 \text{ g L}^{-1}$) instead, which shows lower uptake rates with water addition for the Kangerlussaq soils.

“Moreover, the uptake coefficients of VOCs did not consistently become more negative (i.e., indicating higher uptake potentials) after the water addition. For instance, VOCs such as benzyl alcohol (solubility in water at 20°C : 40 g L^{-1}), furfural (83 g L^{-1}), and acetaldehyde (miscible) showed lower uptake potentials after water addition for the Kangerlussuaq soils, despite being water-soluble, which suggests that physical dissolution alone was not the sole driver of the observed VOC uptake in our study.”

(Lines 189-195)

4. Section 3.5: VOC uptake coefficients: What is the unit for the uptake coefficients? Please include the unit in Table 3.

Response: The unit of the uptake coefficients is $\text{nmol g (soil)}^{-1} \text{ hr}^{-1} \text{ ppb}^{-1}$, or $\text{mol g (soil)}^{-1} \text{ hr}^{-1}$, in our case. We have included this information in the table as well as in the methods section.

5. On line 496: You mention that uptake coefficients can be used for modeling. Could you elaborate on how this would be done? For instance, would it require knowledge of permafrost emissions without an active soil layer? Is such information already available? Would you also need to model air concentrations within the soil? A brief explanation would improve clarity here.

Response: We thank the reviewer for this insightful comment. Yes, using our uptake coefficients to model the uptake rates of different VOCs would require their air concentrations in the ambient atmosphere or within soil pores. For instance, if these concentrations are obtained through in situ observations or modeling predictions, the uptake rates can be approximated by multiplying them with the corresponding coefficients. In the revision, we have added an explanation on how these coefficients may be applied to model the uptake rates of VOCs.

“Specifically, incorporating the uptake kinetics of VOCs, as informed by the uptake coefficients calculated in this study, into modeling frameworks is strongly recommended. For instance, the potential uptake rates of different VOCs can be approximated by multiplying the coefficients with their concentrations, which could be obtained from in situ observational studies or modeling predictions, in either the ambient air or soil pores.” **(Lines 277-282)**